# Elemental and Speciation Analyses of Different Brands of Yerba Mate (*Ilex paraguariensis*)

**DOI:** 10.3390/foods10122925

**Published:** 2021-11-26

**Authors:** Jędrzej Proch, Aleksandra Orłowska, Przemysław Niedzielski

**Affiliations:** Department of Analytical Chemistry, Faculty of Chemistry, Adam Mickiewicz University, Uniwersytetu Poznańskiego 8, 61-614 Poznań, Poland; jedrzej.proch@amu.edu.pl (J.P.); aleksandra.orlowska@amu.edu.pl (A.O.)

**Keywords:** yerba mate *(Ilex paraguariensis)*, speciation analysis, essential trace elements, potentially toxic elements, hyphenated systems, ICP OES

## Abstract

In this work, a methodology for determination of As(III), As(V), dimethylarsinic acid (DMA), Fe(II) and Fe(III) in fifty-eight samples (forty-nine products of thirteen brands from three countries) commercial yerba mate *(Ilex paraguariensis)* was performed. The hyphenated high performance liquid chromatography inductively coupled plasma optical emission spectrometry (HPLC-ICP OES) technique was used. Arsenic was determined below the quantification limit in 38 samples of yerba mate. As(III) was found at the level 0.09 and 0.08 mg kg^−1^. The As(V) content was in the range: 0.21 to 0.28 mg kg^−1^. The content of DMA was found the highest of the three arsenic species in the range: 0.21 to 0.47 mg kg^−1^. The content of Fe(II) and Fe(III) was found in the range: 0.61 to 15.4 mg kg^−1^ and 0.66 to 43.1 mg kg^−1^, respectively and the dominance of Fe(III) was observed. Moreover, total and extractable content of 16 elements were determined. The results have been subjected to statistical analysis in order to establish relationships between samples of the same origin (country), kind (type) and composition (purity).

## 1. Introduction

Yerba mate (*Ilex paraguariensis* St. Hil.) is a native South American tree. It is an important commercial product, consumed in the largest quantities in Brazil and Uruguay, while Argentina is the largest exporter [1]. The consumption of yerba mate has expanded to different countries e.g., Spain, France, Italy, Germany, Korea, Japan, Syria, Russia, United States and Australia [2]. Yerba mate can accumulate both essential trace and potentially toxic elements (PTEs) [1] and their content depends on certain factors, such as soil type, exposure of plants to pollution, even harvest season [3]. High tolerance to metals or metalloids has evolved in a number of plant species. Tolerant plants are often excluders, limiting the entry and translocation, or rarely hyperaccumulators combines extremely high tolerance to, and foliar accumulation of, trace elements [4].

Nowadays (2021), the determination of the total concentration of trace elements in yerba mate or its tea is not enough to evaluate the nutritional or toxic potential of the product. In infusions and decoctions, metals and metalloids may exist as either simple or complexed ions. This fact can affect the bioavailability of elements by humans [5]. What is more, nutritional and toxic potential depends on the element species, therefore a speciation analysis of elements, such as As, Cr, Fe, Se is crucial. The toxicity of elements forms is significantly different, and the greatest obstacle to the efficient determination of these forms is the ease of converting from one form to another [6,7]. In opposite to arsenic, iron (Fe) is an essential trace and the fourth most abundant element in the Earth’s crust. Two main species, Fe(II) and Fe(III), are thermodynamically stable and kinetically reactive, however the role and the demand of these forms in living organisms are different [8]. Due to ambiguous classification of this material in the literature (as tea [9], laboratory plant [10] or wild–growing plant [11]), yerba mate *(Ilex paraguariensis)* is widely used by authors as an application material for new analytical methods [1,5,12,13,14,15,16,17]. It is surprising that the content of essential trace and potentially toxic elements in yerba mate have been obtained by ICP OES [18], ICP MS [1,19] or both ICP OES and ICP MS [11,20], excluding any speciation studies. Admittedly, some methods for determining the species of selenium [14], arsenic [12], iron [13], as well as iron and arsenic [17] in several samples of yerba mate have already been presented. However, speciation studies have not yet been carried out on a larger number of samples.

In this study, to investigate the speciation of arsenic and iron in yerba mate *(Ilex paraguariensis)*, 58 samples were collected from the Polish market. Additionally, the determination of selected essential trace and potentially toxic elements (PTEs), i.e., Al, As, Cd, Co, Cr, Cu, Fe, Hg, Li, Mn, Mo, Ni, Pb, Sb, Se and Zn, was performed using ICP OES. The determination of As(III), As(V) and DMA as well as Fe(II) and Fe(III) in yerba mate is a novel, although it is an enlargement of the preliminary studies, conducted for the first usage of HPLC–HG–ICP OES with MSIS as an interface [12], the first comparison of HPLC–MIP OES and HPLC–ICP OES [13], and the first combination of two HPLC systems with ICP OES through MSIS (2 HPLC-MSIS-ICP OES) [17].

## 2. Materials and Methods

### 2.1. Samples Collecting

Fifty–eight samples of forty–nine yerba mate products from thirteen brands were bought from legal stores located in Poland. Samples were originated from Argentina (*n* = 15), Brazil (*n* = 23), and Paraguay (*n* = 18), however most of samples (*n* = 33) were repackaged in Poland (and distributed under a Polish trademark). According to kind (type), 30 samples were “con palo” (a mixture of 70% leaves and 30% stalks) and 26 samples were “despalada” (a mixture of 90% leaves and 10% stalks). According to composition (purity), 34 samples contained some additives (Table 1) and 24 samples were pure yerba mate. Additionally, nine products (A1, B5, B7, D1, D2, D3, D4, F3, H1) were collected twice (as a 50 g test sample and a 500 g pack). Full details were shown in Table 1.

### 2.2. Gases and Reagents

High-pure argon (N—5.0, purity 99.999%), obtained from Linde, Poland, was employed as a plasma gas. All solutions were prepared using deionized water (≥18 MΩ cm resistivity, water purification system Milli-Q (Merck Millipore, Darmstadt, Germany)). Standard solutions were prepared from commercially available ICP calibration standards (Romil, Cambridge, UK). Disodium hydroarsenate hepahydrate, sodium arsenite and cacodylic acid were collected from Sigma-Aldrich (Saint Louis, MO, USA). Ferric ammonium sulfate dodecahydrate and ferrous ammonium sulfate hexahydrate were obtained from Acros-Thermo Fisher Scientific (Geel, Belgium). Standard solutions of iron and arsenic (1000 mg L^−1^) were prepared by dissolving appropriate amounts of chemical compound in water. Less concentrated standard solutions obtained by dilution of the stock solutions were prepared daily. 1 mol L^−1^ orthophosphoric acid was prepared from 85% H_3_PO_4_ (POCH™, Avantor®, Gliwice, Poland). 65% nitric acid (HNO_3_) was obtained from Merck.

A phosphate buffer was prepared by mixing disodium hydrophosphate (Na_2_HPO_4_) and potassium dihydrophosphate (KH_2_PO_4_·2H_2_O) obtained from Merck. Appropriate amounts of powder reagents were dissolved and mixed to obtain the mobile phase solution, 25 mmol L^−1^ KH_2_PO_4_·2H_2_O and 2.5 mmol L^−1^ Na_2_HPO_4_ (pH adjusted to 6.0 ± 0.2). 1.0% (*w*/*v*) sodium tetrahydroborate (NaBH_4_), was prepared daily, by dissolving appropriate amounts of powdered NaBH_4_ (Sigma-Aldrich) in water and stabilized with 0.1% (*w*/*v*) NaOH (Merck). 5 mol L^−1^ hydrochloric acid was prepared from 30% HCl (Suprapur^®^, Merck). A PDCA eluent was prepared by mixing appropriate volumes of pyridine-2,6-dicarboxylic acid (PDCA) and formic acid (HCOOH) (Sigma-Aldrich), then appropriate amounts of potassium hydroxide (KOH) and potassium sulfate (K_2_SO_4_) (Merck) were dissolved in deionized water. Molar concentrations of PDCA, K_2_SO_4_ KOH, and HCOOH were 7.0, 5.6, 66 and 74 mmol L^−1^ respectively (pH 4.2 ± 0.2). 100 mmol L^−1^ sodium sulfite (Merck) was obtained by dissolving appropriate amounts of powder (Na_2_SO_3_) in water and the solution was used to the periodic column conditioning.

### 2.3. Sample Preparation

Samples were homogenized using agate laboratory grinder (Pulverisette, Fritsch GmbH, Idar-Oberstein, Germany). The procedure of ultrasound-assisted extraction was repeated after previous studies [12,13,17]. Accurately weighted 1.00 (±0.01) g dry samples were placed in a polyethylene test tube. Then 8.0 mL of 1 mol L^−1^ orthophosphoric acid (H_3_PO_4_) was added and the ultrasound-assisted extraction was conducted for 30 min at ambient temperature. Then samples were filtered through a paper filter, washed previously by 200 mL of water and 20 mL of phosphate buffer. Sample solutions were neutralized with a few drops of 15 mol L^−1^ NaOH to obtain pH value in the range of 6.0 to 6.5 and filled up to the final volume of 10 mL. Prepared extracts were tested using ICP OES (extractable content), HPLC-HG-ICP OES (arsenic speciation analysis) and HPLC-ICP OES (iron speciation analysis). Samples were analyzed daily or stored at −30 °C in the laboratory (no longer than a week). Additionally, accurately weighted, 0.500 (± 0.001) g of dry samples, were digested with 5.0 mL of 65% HNO_3_ in closed Teflon^®^ containers (55 mL) using the microwave digestion system, Mars 6 (CEM, Matthews, NC, USA). The process was carried out in three stage: (1) ramping the temperature (20 min), (2) holding at 180 °C (20 min), (3) cooling (20 min). After digestion, samples were diluted with deionized water to a final volume of 10 mL. Digested samples were analyzed using ICP OES (total content of elements).

### 2.4. Arsenic Speciation Studies Using HPLC-HG-ICP OES

The hyphenated technique, HPLC-HG-ICP OES with multi-mode sample introduction system (MSIS, Agilent, Santa Clara, CA, USA) as an interface was previously described in details [12] and it was used to determine As(III), As(V) and DMA. Full experimental conditions of HPLC-HG-ICP OES were summarized in Table 2. The HPLC system was constructed from a HPLC pump, Shimadzu LC-10AT (Shimadzu, Kyoto, Japan) and an anion-exchange HPLC column, Supelco LC-SAX1, 250 mm × 4.6 mm i.d., resin particle size 5 µm (Supelco, Bellefonte, PA, USA). The outlet of the HPLC column was directly connected (avoiding the peristaltic pump) with the tube which supplied HCl using T-shape connecter. The hydride generation is possible due to inlets located vertically in the center of the MSIS unit. The upper inlet was used to provide NaBH_4_ solution. The lower inlet provided an eluent with sample solution and HCl, HG reagents were mixed at the top of the inlets, volatile hydrides were formed and transported into ICP torch by argon (introduced by nebulizer gas inlet). A nebulizer sample inlet stayed blocked. The excess liquid was carried from the chamber using a peristaltic pump. The detector of the hyphenated technique was Agilent 5110 ICP-OES (Agilent). The sample was analyzed three times (*n* = 3). Instrument detection limits (DLs) were calculated using the background equivalent concentration (BEC) and the signal-to-background ratio (SBR). The BEC was calculated as BEC = C_standard_/SBR, where C_standard_ is the concentration of reference standard solution; SBR = (I_standard_ − I_blank_) / I_blank_; and I_blank_ and I_standard_ are the emission intensities [cps] for the blank and the reference standard solution. The DL was calculated (DL = 3 × RSD_blank_ × BEC/100), RSD_blank_ is the relative standard deviation of multiple (*n* = 10) blank measurements. Instrument quantification limits (QLs, calculated as 3.3 × DL), were presented in Table 3 with the method quantification limits, which include a sample preparation. The standard addition method was used for traceability measurements. Recoveries were in the acceptable range (80–120%) and full results were presented in Appendix A. The uncertainty estimated for complete analytical process (with sample preparation step) was below 20%

### 2.5. Iron Speciation Studies Using HPLC-ICP OES

The hyphenated technique, HPLC-ICP OES was previously described in details [13] and it was used to determine Fe(II) and Fe(III). The HPLC system was made up of a Shimadzu LC-10AT HPLC pump and a Dionex IonPac CS5A cation-exchange HPLC column (250 mm × 4.0 mm i.d., resin particle size 5 µm, Thermo Fisher Scientific, Waltham, MA, USA). The outlet of the cation-exchange HPLC column was directly connected (avoiding the peristaltic pump) with the nebulizer sample inlets. Peristaltic pump of ICP-OES system was used to drain the waste only. MSIS was applied as a conventional cyclonic spray chamber with OneNeb nebulizer (both Agilent). The detector of the hyphenated technique was Agilent 5110 ICP-OES (Agilent). The operating conditions were placed in Table 2. The sample was analyzed 3 times (*n* = 3). All DLs and QLs were calculated as described in the Section 2.4 and the results were presented in Table 3. The standard addition method was used for accuracy studies. Recoveries were in the acceptable range (80–120%) and full results were presented in Appendix A. The uncertainty for the complete analytical process (including a sample preparation) was at the level of 20%.

### 2.6. ICP OES Determination of Selected Elements Content

An inductively coupled plasma optical emission spectrometer (Agilent 5110 ICP-OES) was used to determine selected elements (Al, As, Cd, Co, Cr, Cu, Fe, Hg, Li, Mn, Mo, Ni, Pb, Sb, Se, Zn) in acid digests (total content) and extracts (extractable content). Synchronous vertical dual view (SVDV), which allows the axial and radial view analysis simultaneously, was used (viewing height for radial plasma observation was 8 mm). Grading fixed optic was thermostated at 35 °C and a detector, VistaChip II with the charge coupled device (CCD) was cooled to −40 °C using a triple Peltier system. The signal accusation time was 5 s for three replicates. The operating conditions were summarized in Table 2. The sample was analyzed three times (*n* = 3). All DLs and QLs were calculated as described in the Section 2.4 and the results are presented in Table 3. The standard addition method was used for traceability measurements. Recoveries were in the acceptable range (80–120%) and full results were presented in Appendix A. The uncertainty for the complete analytical process with sample preparation step was estimated at the level of 20%.

### 2.7. Statistical Analysis

Statistical analyses were performed using computer software Statistica 13.3 (StatSoft, TIBCO Software Inc., Palo Alto, CA, USA). The distribution of the data was studied by the Kolmogorov-Smirnov, Lilliefors and Shapiro-Wilk tests. Given that the data did not follow a normal distribution, the Spearman’s rank correlation coefficient, which is a nonparametric test, was used. The multidimensional statistical analysis (principal components analysis, PCA) was provided for the results of ICP OES analysis to indicate the individual differences in the elemental composition of the yerba mate samples. The probability value, *p* = 0.05, was applied for all statistical tests.

## 3. Results and Discussion

### 3.1. As and Fe Speciation Studies

When examining the composition of yerba mate in terms of the content of selected forms of arsenic and iron, two methods of sample preparation were used prior to the ICP OES analysis: microwave-assisted digestion to determine As(total) and Fe(total), and ultrasound-assisted extraction to determine As and Fe extractable with phosphoric acid, i.e., As(ext) and Fe(ext). Then extracted samples were analyzed parallel by HPLC-HG-ICP OES determining the content of As(III), As(V) and DMA and HPLC-ICP OES determining the content of Fe(II) and Fe(III). The differences between the extractable content and the sum of arsenic or iron species were described as undefined As or Fe extractable forms, i.e., As(und-ext) and Fe(und-ext). Moreover, the differences between the total content and the extractable content were described as a non-extractable fraction of As and Fe, i.e., As(non-ext) and Fe(non-ext). Results of arsenic and iron speciation studies were presented in Figure 1 as (a) the content of each fraction (mg kg^−1^) and (b) the percentage of each fraction in total content (as 100%).

Arsenic was determined below the quantification limit (BQL) in 38 of the yerba mate samples. As(total) and As(ext) were found above quantification limit (AQL) in 20 and eight samples, respectively. As(ext) was determined BQL in 12 of 20 samples, therefore the content As(non-ext) was equal to As(total). In turn, As(III), As(V) and DMA were detected AQL only in two, four and four samples, respectively. Three arsenic species were only found AQL in one sample (no. 4), which was pure despalada from Argentina. As(III) was found at the similar level, 0.09 and 0.08 mg kg^−1^, in samples nos. 4 and 8, respectively. These samples also contained inorganic As (the sum of As(III) and As(V)) at the highest level, i.e., 0.36 mg kg^−1^. The As(V) content was in the narrow range, from 0.21 (no. 44) to 0.28 mg kg^−1^ (no. 8). The content of DMA was found in the range from 0.21 (no. 58) to 0.47 mg kg^−1^ (no. 12) and it was the highest of the three arsenic species. On one hand, the sum of three arsenic species represented the majority (80–98%) of As(ext), except samples nos. 13 and 18, containing only undefined As extractable forms (As(und-ext)) at relatively low levels, 0.28 and 0.24 mg kg^−1^, respectively. On the other hand, the content of As(ext) was 29% (as median) of As(total).

The content of As(total) and As(ext) exceeded the recommended limit (0.60 mg kg^−1^), established by legislation [21], in 16 and two samples (nos. 4 and 53), respectively. Surprisingly, the mean value of As(total) (1.35 mg kg^−1^) was 26 [20], 28 [11,19] and 34 times higher [1,22] than in the literature, however it was also found BQL in 38 samples. Additionally, it is difficult to clearly indicate whether the ultrasound-assisted extraction in 1 mol L^−1^ H_3_PO_4_ (at room temperature) allows to extract its higher content than a traditional extraction in hot water (80 ± 10 °C). Opinions are divided in the literature and the following percentage of total content were found in hot water, 18% [22], 48% [20] and 49% [10]. Therefore, the risk of exceeding the recommended level cannot be directly assessed. The determination of three As species has not been performed on yerba mate samples yet, except the introduction as an application material for HPLC-HG-ICP OES [12] and 2 HPLC-MSIS-ICP OES [17], therefore no comparable data are available in the literature. In the first work, As(III), As(V) and DMA were found at lower ranges in six samples: 0.012–0.032 mg kg^−1^, 0.046–0.108 mg kg^−1^ and 0.044–0.097 mg kg^−1^, respectively [12]. In the second work, only As(V) was determined in three samples (0.31–0.33 mg kg^−1^), while As(III) and DMA were found BQL in all samples (*n* = 8) [17]. Although results were rather comparable with those obtained in previous studies, occurring As and its species in yerba mate as well as occasionally the high content, are accidental rather than related to studied factors, i.e., country of origin and packing, kind or purity.

Fe(total) and Fe(ext) were found in the following ranges: 51.1–1059 mg kg^−1^ and 2.92–62.8 mg kg^−1^, respectively, in the whole sample population (*n* = 58). In the case of Fe, it is difficult to clearly indicate whether the ultrasound-assisted extraction in 1 mol L^−1^ H_3_PO_4_ (at room temperature) allows one to extract a higher amount than a traditional extraction in hot water (80 ± 10 °C). Opinions on this in the literature are divided and the following percentages of total content were found in hot water: 1.3% [23], 2% [22], 3% [10] and 15% [20]. However, the percentage of extraction by cold water (1.1%) was similar to hot water (1.3%) [23], therefore a higher percentage of extraction may be obtained with the method proposed in this study, 8% (as median). Ionic Fe(II) and Fe(III) were determined AQL in 26 and 31 samples respectively. The content of Fe(II) and Fe(III) was ranging, 0.61–15.4 mg kg^−1^ and 0.66–43.1 mg kg^−1^, respectively. Although the Fe(III)/Fe(II) ratio was found in wide range (0.3–33), the dominance of Fe(III) was observed (median = 2.7). Higher content of Fe(II) was determined in four samples (nos. 6, 29, 37, 39). The highest content of Fe(II) and Fe(III) was detected in samples nos. 11 and 9, respectively, which were both Paraguayan yerba mate con palo with additives (aromas) and distributed under the same Polish brand (Table 1).

The determination of two Fe species has not been performed on yerba mate samples yet, except the introduction as an application material for HPLC-ICP OES and HPLC-MIP OES [13] and 2 HPLC-MSIS-ICP OES [17], therefore no comparable data are found in the literature. In the first work, higher content of Fe(II) and Fe(III) were found in 2 (15–31 mg kg^−1^) and three samples (16–43 mg kg^−1^) respectively [13]. In the second study, Fe(III) was found in the range from 1.33 to 19.1 mg kg^−1^ in all samples (*n* = 8), while Fe(II) was determined in four samples (3.07–7.05 mg kg^−1^) [17]. The dominance of Fe(III) was observed in mentioned studies what may indicate the implication of the oxidizing conditions of the production process of yerba mate (e.g., roasting) [24]. Based on data from preliminary studies [13,17], it was expected that the content of ionic Fe(II) and Fe(III) in yerba mate is widespread. Surprisingly, only Fe(und-ext) was detected in 27 samples. On the one hand, this may indicate the presence of certain extractable iron complex compounds in yerba mate, which remain stable despite the use of ultrasound-assisted extraction and are inert towards cation-exchange column. On the other hand, these complexes could also be formed with any compounds extractable under these conditions from the sample matrix. Nevertheless, in order to confirm or reject this hypothesis, it would be advisable to continue speciation studies on yerba mate, including by increasing the number of samples as well as analytes.

### 3.2. Total and Extractable Content of Selected Elements

The samples were considered as a whole (*n* = 58) and three groups, depending on the country of origin (Argentina, Brazil or Paraguay), kind (con palo or despalada) or composition (pure yerba mate or those with additives) (Table 1). The total content of selected essential trace and potentially toxic elements in yerba mate *(Ilex paraguariensis)* was presented in Table 4. The Kolmogorov-Smirnov, Lilliefors and Shapiro-Wilk tests indicated that the data did not follow a normal distribution (except Cu, what may be related with using Cu containing chemical products to control fungal diseases). According to this, the Spearman’s rank correlation coefficient was used and results are discussed in Section 3.3.

Nine elements (Al, Cd, Co, Cr, Cu, Fe, Mn, Ni and Zn) were found AQL in all samples. Among the trace elements in yerba mate, the manganese content is indisputably the highest. Generally, the following order of total concentration (as median): Mn > Al > Fe > Zn > Cu > Ni > other elements, was observed in the whole population and every group which was reported before in several studies [11,20,22,23]. The only difference was Fe > Al in Paraguayan samples (Table 4). Other elements, which were found in the whole population (*n* = 58), were arranged in the following order: Cr > Co > Cd. Moreover, the same order was also observed for despalada, yerba mate with additives or those originating from Brazil and Paraguay. However, the different order (Cr > Cd > Co) was found in con palo and pure yerba mate. The order Cr > Cd > Co was also reported in the literature [11,20]. The order Co > Cr > Cd was observed in the case of Argentinian samples (*n* = 15). Seven elements, As, Hg, Li, Mo, Pb, Sb and Se, were found AQL in 20, 27, 42, 54, 15, 50 and 38 samples respectively (Table 5). The order As > Sb > Se > Pb was observed in the whole population and con palo samples. Moreover, the order Sb > Se > As > Pb was found in the case of despalada and Brazilian samples while it was different for Argentinian (As > Sb > Se > Pb), Paraguayan (Sb > As > Se > Pb) samples and pure yerba mate (As > Pb > Sb > Se). Moreover, Pb > Se distinguished pure and Brazilian samples. The content of Hg, Li and Mo was similarly low and the order Hg > Mo > Li was observed in each subgroup and the whole population. It is worth mentioning that the same order, containing all of 16 elements (in total content), was observed in two subgroups (despalada and Brazilian samples). The above compliance is due to the fact that all Brazilian samples (*n* = 23) are despalada (*n* = 26).

The extractable content of selected essential trace and potentially toxic elements in 58 samples of yerba mate *(Ilex paraguariensis)* is presented in Table 5. In contrast to total content, only five elements were found AQL in all samples (Al, Cu, Fe, Mn, and Zn). Moreover, Ni was detected BQL only in one sample (no. 1), which was a pure yerba mate con palo from Argentina. In turn, Cd, Co and Sb were determined in 48, 47 and 48 extracts, respectively. It is worth noting that other studied elements were found AQL only in one (Hg), two (Li), three (Mo), seven (Pb and Se) and eight samples (As). However, Cr extractable content was significantly higher than its total content. The increase in the results was repeatable and it seemed to be caused by unexpected signal interferences related to the selection of the sample preparation method (ultrasonic extraction with diluted phosphoric acid) or a peculiarity of the sample matrix. Nevertheless, this problem would require further clarification, therefore the results for the Cr extractable content was rejected.

The following order of extractable content was observed (as median) in the whole population and each subgroup: Mn > Al > Zn > Fe > Cu > Ni > other elements. Considering only elements, whose results were BQL for <20% of all samples (i.e., Al, Cd, Co, Cu, Fe, Mn, Sb, and Zn), the order for each subgroup as well as the whole population was identical: Mn > Al > Zn > Fe > Cu > Ni > Sb > Co > Cd. In the case of other elements (As, Pb and Se), the order Se > Pb > As was found in the whole population and four subgroups (con palo, despalada, yerba mate with additives and those originating from Brazil). The order Pb >Se > As was observed for pure yerba mate (*n* = 24).

For the whole sample population (*n* = 58), the percentages of extraction (as median) were arranged in the ascending order: Fe (8%), Cd (17%), Mn (23%), Ni (27%), Al (28%), As (29%), Li (34%), Zn (35%), Sb (36%), Co (37%), Cu (39%), Pb (56%), Mo (57%), Se (61%), and Hg (84%), however Hg(ext) was only determined AQL in one sample (no. 53, Argentinian yerba mate con palo with additives). Comparing to other studies, where the percentages were obtained after hot water extraction [10,20,22,23,25], the percentages in this study were both lower (Co) and higher (Al and Se). In the case of other elements, it is difficult to clearly indicate whether the ultrasound-assisted extraction in 1 mol L^−1^ H_3_PO_4_ (at room temperature) allows to extract its higher content than a traditional extraction in hot water (80 ± 10 °C). According to the studies mentioned above, As, Cu, Li, Mn and Ni were usually better extracted using hot water and Cd, Fe, Mo, Pb and Zn were rather better extracted with diluted phosphoric acid [10,20,22,23,25].

In the case of tea or yerba mate, the maximum limits of total concentrations of As, Cd and Pb were established by legislation (0.6, 0.4, and 0.6 mg kg^−1^, respectively) [21], however, the results for arsenic have already been discussed in Section 3.1. The content of Pb(total) and Pb(ext) exceeded the recommended limit (0.6 mg kg^−1^) in eight and three samples (nos. 16, 19 and 50), respectively. The mean value of Pb(total) (0.86 mg kg^−1^) was slight higher than in the literature [10,20,22,23], however samples exceeding the Pb limit were also reported before [1,15,19,26]. Nevertheless, Pb(total) was found BQL in 43 samples. In turn, Cd(total) was determined in all samples (*n* = 58), while Cd(ext) was found AQL in 48 samples. The content of Cd(total) and Cd(ext) exceeded the recommended limit (0.4 mg kg^−1^) in 30 and six samples (nos. 4, 8, 43, 46, 50, 53), respectively. Moreover, the mean value of Cd(total) (also the median, 0.41 mg kg^−1^) exceeded the limit. These results corresponds to several studies which indicate that Cd was frequently found above 0.4 mg kg^−1^ [1,15,19,20,22,23,26,27]. Based on these findings, these concentrations may be considered natural and established limits should be revised, especially for Cd and Pb [26,27].

For easier evaluation of the results obtained in this study, the total content of elements in yerba mate samples was compared with the literature data (Table 6). In addition, the results were also divided into groups (origin, kind and composition), as pure yerba mate samples and those with additives have not been compared previously in terms of the elemental composition.

According to the latest literature, the total content of all selected elements was variable, however the results were generally in the same order of magnitude, except for As and Li (Table 6). Generally, most of determined elements had slightly higher contents than in the literature and the exceptions were Cu and Li. Significant excess levels of selected elements in comparison with the literature may be caused by dust residue deposition, which increased Fe, Ti, As, Pb, Li, Mo, and V concentrations of foliar tissue, what was confirmed in the washing process [26]. As samples were not washed prior to their homogenization, it explains higher content of As, Fe, Mn, Pb, Mo, Ni and Sb found in our study than in the literature (Table 6). It was reported that field handling, transportation, and loading procedures were likely routes of soil contamination of commercial products [26].

Nevertheless, some observations in accordance with studied groups were similar to the literature data. According to kind (con palo or despalada), obtained results for total content of Cu, Ni, Pb and Zn corresponded to those obtained by Baran et al. [25], however the results for Cd, Fe and Mn showed an opposite trend. Authors similarly reported the higher content of Cr and Zn and lower of Mn in Paraguayan yerba mate than those originating from Argentina [25]. According to origin, the highest content of Mn and Co in Argentinean samples, the lowest content of Mn and Co in Paraguayan samples and the lowest content of Fe, Ni and Zn in Brazilian samples were reported by Pozebon et al. [20]. According to the composition, samples of pure yerba mate contained higher content of Mn, Fe, Al, Ni, As, Co and Mo than yerba mate with additives (*n* = 34), which contained more Zn, Cu, Sb, Se, Pb and Hg. Nevertheless, no significant data have been found in the literature evaluating the influence of additives on the elemental composition of yerba mate. It is worth adding that the results of the total elemental content in the context of the whole population of yerba mate samples were similar to those in the literature, except for the aforementioned exceptions [11,23].

### 3.3. Spearman’s Rank Correlation Test

The Spearman’s rank correlation coefficient (r_s_) was used to describe the pairwise associations between total and extractable content of elements in yerba mate samples. The plot of the Spearman’s correlation matrix of elements was presented in Appendix A.

For total content of 16 elements, there were no strong positive (r_s_ ≥ 0.7) nor strong negative correlations (r_s_ ≤ −0.7). Moderate positive correlations (0.4 ≤ r_s_ < 0.7) were observed between Zn/Cd, Mn/Co, Zn/Fe, Li/Al, Ni/Mn, Pb/Cu, Cu/Al, Pb/Al, Pb/Li, Mn/Cr, Ni/Co, Li/Fe, Zn/Li and Fe/Al. Other positive correlations, which were statistically significant (*p* < 0.05), were weak (r_s_ < 0.4). In turn, moderate negative correlation (−0.7 ≤ r_s_ < −0.4) was observed between the pairs: Li/Co, Cu/Co, Pb/Co, Co/Al and Mn/Li. Other negative correlations, which were statistically significant, were weak (r_s_ > −0.4). In the literature, positive correlations were also reported for Ni/Mn (0.78), Pb/Cu (0.71) [25] and Al/Pb (0.55) [26]. For extractable content of 15 elements (without Cr), there were two pairs of elements almost fully positive correlated (r_s_ ≥ 0.9): Mn/Al and Cu/Al. Strong positive correlations (0.7 ≤ r_s_ < 0.9) were observed between Fe/Cu, Ni/Al, Ni/Mn, Fe/Al, Ni/Fe, Ni/Cu, Mn/Cu, Fe/Cd and Zn/Fe, while moderate positive correlations (0.4 ≤ r_s_ < 0.7) were observed between the following pairs: Mn/Fe, Cu/Cd, Zn/Cu, Zn/Cd, Cd/Al, Ni/Cd, Zn/Ni, Zn/Al, Zn/Mn and Mn/Cd. All statistically significant negative correlations (Li/Co and Pb/Mn) were weak (r_s_ > −0.4). The r_s_ coefficient was also calculated between total and extractable content, however there were no strong positive (r_s_ ≥ 0.7) nor strong and moderate negative correlations (r_s_ ≤ −0.4). Moderate positive correlations (0.4 ≤ r_s_ < 0.7) were observed for total and extractable of Pb, As, Sb, Mn, Co and Ni (r_s_ = 0.39). Moreover, positive correlation for total and water extractable content of Mn, Ni and Co were also reported [22]. It is presumed that the higher concentrations as well as significant correlations of the total and extracted content, are related to the origin of these elements, which is dust residue deposition. It was confirmed that the higher contents of some elements in foliar tissue of yerba mate *(Ilex paraguariensis)* came from surface deposition of soil dust [26]. In the case of other elements, the r_s_ coefficient was not significant statistically (*p* < 0.05) between their total and extractable content. In turn, moderate positive correlation was also observed in the following pairs: Mn(ext)/Ni(total), Ni(ext)/Mn(total) and Zn(ext)/Cd(total), while weak positive correlation for Co(ext)/Cr(total), Al(ext)/Mn(total), Fe(ext)/As(total), As(ext)/Fe(total), Al(ext)/Ni(total), Sb(ext)/Mn(total), Sb(ext)/Cd(total), Co(ext)/Mn(total), Cd(ext)/Se(total), Li(ext)/Se(total), Cu(ext)/Mn(total), As(ext)/Sb(total). There were also observed weak negative correlations between the pairs: Mn(ext)/Li(total), Mn(ext)/Zn(total), Pb(ext)/Mn(total), and Cu(ext)/Li(total) (Appendix A). Significantly high correlations between Zn and Cd may be associated with their chemical similarities [23], while other correlations (e.g., Fe/As) may indicate that these elements came from the common source (e.g., soil dust) [26].

### 3.4. Principal Component Analysis (PCA)

The results, obtained in the ICP OES analysis, were subjected to analysis by principal component analysis (PCA). Two components described 99.6% variability of the results of total concentration of 16 elements (*n* = 928 of single results for 58 digested samples of yerba mate) and 99.8% variability of the results of extractable concentration of 15 elements (*n* = 870 of single results for 58 samples of yerba mate extracted with H_3_PO_4_ solution). In both cases, all samples were cumulated in one group (at the point +1.0 of the PC1 axis and it was distributed along the PC2 axis, from −0.2 to 0.3). According to this, it is clearly indicated that samples were similar in the elemental composition (both in accordance with total and extractable content). In the case of total content of 16 elements, 11 samples could be distinguished from the group, nos. 7, 14, 19, 29, 32, 33, 35, 36, 39, 43, 53, and most of them were paired as followed: 29 + 35, 32 + 33 (both pairs were Brand D), 39 + 43 (both samples were Brand E) and 7 + 53. What is more, samples nos. 32 and 33 were the same product (D3), collected both as a 500 g pack (no. 32) and a 50 g test sample (no. 33). The differentiation in terms of the total content of elements may be due to the additives (except samples nos. 7, 29 and 36 were pure yerba mate). Moreover, half of the Brand D samples (nos. 29, 32, 33 and 35) is also distinct. In the case of extractable content of 15 elements, only five samples could be distinguished from the group, nos. 9, 19, 29, 34, 35. The only pair (34 + 35) was the same product (D4), collected both as a 500 g pack (no. 34) and a 50 g test sample (no. 35). As in the case of the total content, differentiations may be related to the additives (except sample no. 29). Only samples nos. 19, 29 and 35 were differed from the group in both contents, however their only common factor is being test samples.

The PCA for total and extractable content did not show any significant individual differences between the studied samples. Due to this, the percentages of extraction were additionally subjected to analysis by the PCA, however the data were reduced to nine elements only (*n* = 490 of single results for 58 samples), whose results were BQL for <20% of all samples. Reduced data represented 93.9% of all results. The 70.4% variability of results was described by two components (Figure 2). For easy identification, samples were colored and divided in accordance with origin, type, composition and brand.

Two groups of samples could be formed (Figure 2). According to the country of origin, the Argentinian samples were distributed mainly in group 1, however samples nos. 3, 47, 50, 52 and 53 were in group 2. A substantial similarity between samples was visible for two brands: Brand A (except for sample no. 3) and Brand F (except for sample no. 47). In turn, the Brazilian samples were distributed between both groups (7 and 11 samples in group 1 and 2, respectively), however two samples (nos. 19 and 26) were significantly separated. A substantial difference between the two brands was visible in the case of the Brazilian yerba mate. Brand E samples were on the positive part of the PC2 axis (except for sample no. 36), while the Brand C samples were usually on the negative part of the PC2 axis (except for samples nos. 19 and 26). The Paraguayan samples were generally in group 2, however samples nos. 7, 10, 11 were out of the group. Moreover, the Brand D samples were on the negative part of the PC2 axis (except for sample no. 28, which was separated).

According to the kind of yerba mate, the con palo samples were distributed between both groups in the ratio 1:2 (9 and 18 samples in group 1 and 2, respectively) while despalada samples were distributed proportionally between both groups (10 and 11 samples in group 1 and 2, respectively). According to the composition (purity), samples were also distributed between both groups. Pure yerba mate samples were distributed proportionally between both groups, with 11 and 10 samples in group 1 and 2, respectively. In turn, nine and 20 samples of yerba mate with additives were distributed in group 1 and 2, respectively. Therefore, the PCA for the extraction percentages did not indicate any significant order in terms of the type (type) or composition (purity) of the yerba mate samples. Nevertheless, selected products, which were collected twice (a 500 g pack and a 50 g test samples), were generally distributed in the same group, except samples of B5 (nos. 11 and 12), and H1 (nos. 51 and 52).

The PCA results, such as the lack of a clear grouping of samples according to the country of origin, type (kind) or composition (purity), are partially confirmed by the latest literature data. Although the difference in elemental composition of yerba mate, indicated by the multidimensional statistical analysis, was defined as a geographical factor [20,22,28], it should be linked with the soil parent material where plants were cultivated [26]. This observation was also reported when the Brazilian samples from three different states were studied [22].

## 4. Conclusions

New methods of single-element speciation analysis have been applied for the first time to 58 commercial samples of yerba mate. As expected, the results of arsenic and iron species were comparable with those obtained for several samples in our preliminary studies [12,13,17]. By comparing the content of selected essential trace and potentially toxic elements obtained with two different procedures of sample preparation (digestion, extraction), it was shown that non-extractable content predominated in yerba mate, especially for iron (approx. 92%). According to this, the determination of the arsenic and iron species as well as extractable content must be assessed in the context of the total content of these elements since the elements were built into the plant tissues during the growth stage. The occurrence of selected As and Fe species in yerba mate as well as occasionally the high content, are accidental rather than related to studied factors, i.e., country of origin, kind, composition or packing. What is more, no significant differences were noticed between the yerba mate products packaged in Poland and those packaged in South American countries. Probably the soil which was used for the cultivation of *Ilex paraguariensis*, is the main factor influencing the elemental composition, and not the country of origin as previously thought. In the future perspective, new methods of multi-element speciation analysis as well as an appropriate preparation procedure of this sample matrix should be developed to obtain more information about the origin, type and composition of yerba mate.

## Figures and Tables

**Figure 1 foods-10-02925-f001:**
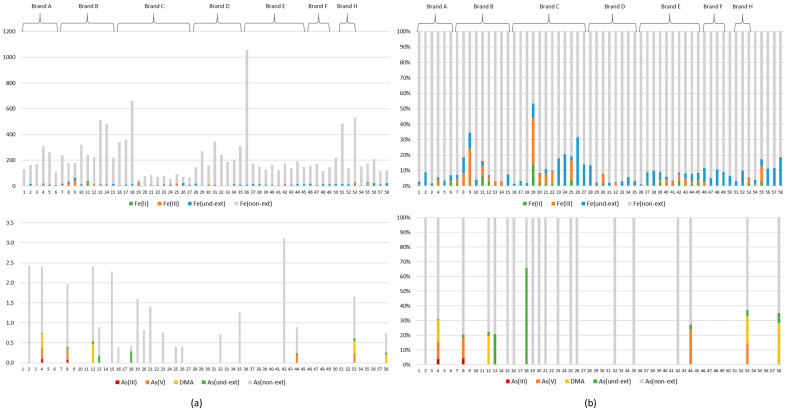
Results of iron and arsenic speciation analysis of yerba mate: (**a**) the concentration of each fraction in total content (as a sum, mg kg^−1^) and (**b**) the percentage of each fraction in total content (as 100%). Captions: As(und-ext), Fe(und-ext)—undefined extractable fraction of arsenic and iron (i.e., the difference between extractable content and the sum of species content); As(non-ext), Fe(non-ext)—non-extractable fraction of arsenic or iron (i.e., the difference between total and extractable content).

**Figure 2 foods-10-02925-f002:**
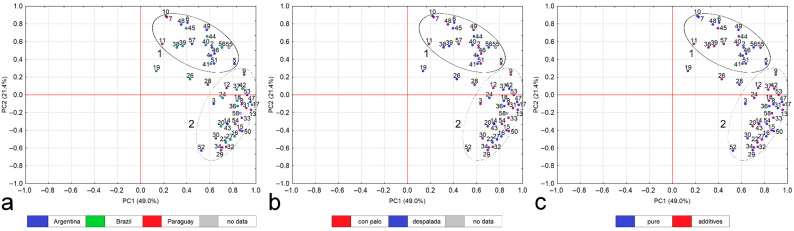
Results of Principle Components Analysis for the extractable percentages of selected elements (Al, Cd, Co, Cu, Fe, Mn, Ni, Sb, Zn) in yerba mate samples (*n* = 58) in accordance with (**a**) origin, (**b**) kind, and (**c**) composition. The 70.4% variability of the results was described by two components (PC1, PC2).

**Table 1 foods-10-02925-t001:** Details of samples’ characteristics.

Sample No.	Product Code	Country of Origin	Type (Kind)	Additives	Packing Type (Weight)
1	A1	Argentina	Con palo	–	a pack (500 g)
2	A1	Argentina	Con palo	–	a test sample (50 g)
3	A2	Argentina	Con palo	–	a test sample (50 g)
4	A3	Argentina	Despalada	–	a pack (500 g)
5	A4	Argentina	Con palo	–	a pack (500 g)
6	A5	Argentina	Despalada	–	a pack (500 g)
7	B1	Paraguay *	Con palo	–	a test sample (50 g)
8	B2	Paraguay *	Con palo	–	a test sample (50 g)
9	B3	Paraguay *	Con palo	aromas	a test sample (50 g)
10	B4	Paraguay *	Despalada	–	a test sample (50 g)
11	B5	Paraguay *	Con palo	aromas	a pack (500 g)
12	B5	Paraguay *	Con palo	aromas	a test sample (50 g)
13	B6	Paraguay *	Con palo	herbs	a test sample (50 g)
14	B7	Paraguay *	Con palo	herbs	a pack (500 g)
15	B7	Paraguay *	Con palo	herbs	a test sample (50 g)
16	C1	Brazil *	Despalada	–	a test sample (50 g)
17	C2	Brazil *	Despalada	fruits, aromas	a test sample (50 g)
18	C3	Brazil *	Despalada	fruits, herbs, aromas	a test sample (50 g)
19	C4	Brazil *	Despalada	fruits, flowers, herbs	a test sample (50 g)
20	C5	Brazil *	Despalada	herbs, fruits, aromas	a test sample (50 g)
21	C6	Brazil *	Despalada	flowers, herbs, seeds, aromas	a test sample (50 g)
22	C7	Brazil *	Despalada	fruits, flowers, aromas	a test sample (50 g)
23	C8	Brazil *	Despalada	fruits, aromas	a test sample (50 g)
24	C9	Brazil *	Despalada	herbs, fruits, aromas	a test sample (50 g)
25	C10	Brazil *	Despalada	herbs, fruit skin, aromas	a test sample (50 g)
26	C11	Brazil *	Despalada	fruit skin, herbs, aromas	a test sample (50 g)
27	C12	Brazil *	Despalada	fruits, flowers, aromas	a test sample (50 g)
28	D1	Paraguay	Con palo	–	a pack (500 g)
29	D1	Paraguay	Con palo	–	a test sample (50 g)
30	D2	Paraguay	Con palo	–	a pack (500 g)
31	D2	Paraguay	Con palo	–	a test sample (50 g)
32	D3	Paraguay	Con palo	aromas	a pack (500 g)
33	D3	Paraguay	Con palo	aromas	a test sample (50 g)
34	D4	Paraguay	Con palo	herbs	a pack (500 g)
35	D4	Paraguay	Con palo	herbs	a test sample (50 g)
36	E1	Brazil *	Despalada	–	a test sample (50 g)
37	E2	Brazil *	Despalada	fruits, herbs, flowers, aromas	a test sample (50 g)
38	E3	Brazil *	Despalada	herbs, fruit skin, aromas	a test sample (50 g)
39	E4	Brazil *	Despalada	fruits, aromas	a test sample (50 g)
40	E5	Brazil *	Despalada	herbs, fruits, aromas	a test sample (50 g)
41	E6	Brazil *	Despalada	fruits, herbs, aromas	a test sample (50 g)
42	E7	Brazil *	Despalada	fruit skin, aromas	a test sample (50 g)
43	E8	Brazil *	Despalada	fruits, herbs, flowers, aromas	a test sample (50 g)
44	E9	Brazil *	Despalada	fruits, flowers, aromas	a test sample (50 g)
45	E10	Brazil *	Despalada	herbs, flowers, aromas	a test sample (50 g)
46	F1	Argentina	Con palo	–	a pack (500 g)
47	F2	Argentina	Con palo	–	a pack (500 g)
48	F3	Argentina	Con palo	–	a pack (500 g)
49	F3	Argentina	Con palo	–	a test sample (50 g)
50	G1	Argentina	Con palo	–	a test sample (50 g)
51	H1	Argentina	Con palo	–	a test sample (50 g)
52	H1	Argentina	Con palo	–	a pack (500 g)
53	H2	Argentina	Con palo	aromas	a test sample (50 g)
54	I1	Paraguay	Con palo	–	a test sample (50 g)
55	J1	N/D *	N/D	fruits, flowers, herbs	a weighted pack (100 g)
56	K1	Brazil	Despalada	–	a test sample (50 g)
57	L1	Argentina	Con palo	herbs, aromas	a pack (500 g)
58	M1	N/D *	N/D	fruit skin, fruits	a weighted pack (100 g)

The Latin letter means the brand (A–M), and the following Arabic number means the same product; ***** means repacked in Poland and distributed as the Polish trademark; N/D means no data.

**Table 2 foods-10-02925-t002:** Operating conditions of HPLC-HG-ICP OES (arsenic speciation analysis), HPLC-ICP OES (iron speciation analysis) and ICP OES (total and extractable content).

	HPLC-HG-ICP OES	HPLC-ICP OES	ICP OES
**HPLC conditions**			
Pump	Shimadzu LC-10AT	Shimadzu LC-10AT	N/A
Column	Supelco LC-SAX 1 (250 mm × 4.6 mm i.d., 5 µm)	Dionex IonPac CS5A (250 mm × 4.0 mm i.d., 5 µm)	N/A
Mobile phase	Phosphate buffer	PDCA eluent	N/A
composition	2.5 mmol L^−1^ disodium hydrophosphate (Na_2_HPO_4_), 25 mmol L^−1^ potassium dihydrophosphate (KH_2_PO_4_ 2H_2_O)	7.0 mmol L^−1^ pyridine-2,6-dicarboxylic acid (PDCA), 66 mmol L^−1^ potassium hydroxide (KOH), 5.6 mmol L^−1^ potassium sulfate (K_2_SO_4_), 74 mmol L^−1^ formic acid (HCOOH)	N/A
pH	6.0 ± 0.2	4.2 ± 0.2	N/A
flow rate [mL min^−1^]	2.0	2.0	N/A
Injection volume [mL]	0.2	0.2	N/A
Spray chamber type	MSIS (Agilent)	MSIS (Agilent)	Double-pass cyclonic (Agilent)
Work mode	HG	Nebulization	Nebulization
Sample channel	Lower inlet	Nebulizer (OneNeb, Agilent)	Nebulizer (OneNeb, Agilent)
**HG parameters**			
NaBH_4_ concentration [%, *w/v*]	1.0	N/A	N/A
NaOH concentration [%, *v/v*]	0.1	N/A	N/A
HCl concentration [mol L^−1^]	5.0	N/A	N/A
NaBH_4_ flow rate [mL min^−1^]	1.0	N/A	N/A
HCl flow rate [mL min^−1^]	1.0	N/A	N/A
**ICP OES conditions**			
Spectrometer	ICP 5110 Dual-View	ICP 5110 Dual-View	ICP 5110 Dual-View
RF power [kW]	1.45	1.20	1.20
Nebulizer gas flow [L min^−1^]	0.7	0.7	0.7
Plasma gas flow [L min^−1^]	12	12	12
Auxiliary gas flow [L min^−1^]	1.0	1.0	1.0
Torch view	axial	SVDV	SVDV
Analytical wavelengths [nm]	As 188.980	Fe 238.204	Al 396.152, As 188.980, Cd 214.439, Cr 267.716, Co 238.892, Cu 327.395, Fe 238.204, Hg 194.164, Li 670.783, Mn 257.610, Mo 202.032, Ni 231.604, Pb 220.353, Sb 206.834, Se 196.026, Zn 213.857.

MSIS—Multi-mode Sample Introduction System; HG—Hydride generation; SVDV—synchronous vertical dual view.

**Table 3 foods-10-02925-t003:** Instrument and method quantification limits (including a sample preparation procedure).

	Al ^a^	As ^a^	As(III) ^b^	As(V) ^b^	DMA ^b^	Cd ^a^	Co ^a^	Cr ^a^	Cu ^a^	Fe ^a^	Fe(II) ^c^	Fe(III) ^c^	Hg ^a^	Li ^a^	Mn ^a^	Mo ^a^	Ni ^a^	Pb ^a^	Sb ^a^	Se ^a^	Zn ^a^
QL [µg L^−1^]	9.3	18	6.2	21	19	0.5	0.7	0.6	0.5	6.7	36	30	4.0	0.3	5.8	1.5	5.0	12	17	7.0	1.6
QL(ext) [mg kg^−1^]	0.093	0.180	0.062	0.210	0.190	0.005	0.007	*	0.005	0.067	0.360	0.300	0.040	0.003	0.058	0.015	0.050	0.120	0.170	0.070	0.016
QL(total) [mg kg^−1^]	0.190	0.360	x	x	x	0.010	0.014	0.012	0.010	0.134	x	x	0.080	0.006	0.116	0.030	0.100	0.240	0.340	0.140	0.032

**^a^**—determined by ICP OES; ^b^—determined by HPLC-HG-ICP OES; ^c^—determined by HPLC-ICP OES; QL—quantification limit (as 10 standard deviation of the blank); QL(total)—method quantification limit (including microwave-assisted digestion); QL(ext)—method quantification limit (including ultrasound-assisted extraction); x—not determined; *—not determined (due to interferences, details in text).

**Table 4 foods-10-02925-t004:** Total content (mg kg^−1^) of selected essential trace and potentially toxic elements in yerba mate *(Ilex paraguariensis)* in accordance with the whole population, origin, kind (type) and composition (purity).

	Origin			Kind (Type)		Composition (Purity)		Whole Population
	Argentina (*n* = 15)	Brazil (*n* = 23)	Paraguay (*n* = 18)	Con Palo (*n* = 30)	Despalada (*n* = 26)	Pure Mate (*n* = 24)	With Additives (*n* = 34)	(*n* = 58)
Elements	Median {Range} (AQL)	Median {Range} (AQL)	Median {Range} (AQL)	Median {Range} (AQL)	Median {Range} (AQL)	Median {Range} (AQL)	Median {Range} (AQL)	Median {Range} (AQL)
Al	210 {86–337}	220 {97–366}	208 {128–371}	210 {86–371}	220 {97–366}	226 {128–366}	207 {86–371}	215 {86–371}
	(15)	(23)	(18)	(30)	(26)	(24)	(34)	(58)
As	2.40 {1.66–2.43}	0.79 {0.39–3.11}	1.61 {0.71–2.42}	1.81 {0.71–2.43}	0.83 {0.39–3.11}	2.18 {0.39–2.43}	0.89 {0.40–3.11}	1.08 {0.39–3.11}
	(3)	(10)	(6)	(8)	(11)	(4)	(16)	(20)
Cd	0.30 {0.16–0.52}	0.41 {0.25–0.68}	0.43 {0.26–0.64}	0.41 {0.16–0.64}	0.40 {0.24–0.68}	0.41 {0.16–0.62}	0.40 {0.24–0.72}	0.41 {0.16–0.72}
	(15)	(23)	(18)	(30)	(26)	(24)	(34)	(58)
Co	0.79 {0.16–1.68}	0.34 {0.02–0.94}	0.33 {0.17–0.60}	0.41 {0.16–1.68}	0.37 {0.02–0.94}	0.47 {0.16–1.68}	0.34 {0.02–0.94}	0.39 {0.02–1.68}
	(15)	(23)	(18)	(30)	(26)	(24)	(34)	(58)
Cr	0.64 {0.26–2.11}	0.57 {0.01–1.20}	0.52 {0.35–0.79}	0.53 {0.26–2.11}	0.56 {0.01–1.20}	0.59 {0.30–1.20}	0.53 {0.01–2.11}	0.55 {0.01–2.11}
	(15)	(23)	(18)	(30)	(26)	(24)	(34)	(58)
Cu	5.46 {3.01–9.20}	7.14 {4.45–9.50}	6.32 {4.62–10.6}	6.16 {3.95–10.6}	7.06 {3.01–9.50}	5.60 {3.01–9.50}	6.98 {3.95–10.6}	6.75 {3.01–10.6}
	(15)	(23)	(18)	(30)	(26)	(24)	(34)	(58)
Fe	164 {113–532}	141 {51–1059}	233 {145–510}	196 {118–532}	144 {51–1059}	176 {113–1059}	170 {51–660}	173 {51–1059}
	(15)	(23)	(18)	(30)	(26)	(24)	(34)	(58)
Hg	0.29 {0.09–0.59}	0.26 {0.09–1.34}	0.33 {0.03–0.56}	0.33 {0.03–0.59}	0.22 {0.09–1.34}	0.31 {0.03–0.59}	0.30 {0.09–1.34}	0.30 {0.03–1.34}
	(8)	(10)	(8)	(15)	(11)	(12)	(15)	(27)
Li	0.06 {0.01–0.09}	0.06 {0.01–0.18}	0.10 {0.03–0.21}	0.07 {0.01–0.21}	0.06 {0.01–0.18}	0.07 {0.01–0.21}	0.06 {0.01–0.21}	0.06 {0.01–0.21}
	(9)	(14)	(18)	(25)	(16)	(19)	(23)	(42)
Mn	2717 {986–3461}	1518 {800–2321}	1231 {653–1966}	1396 {653–3461}	1650 {800–3265}	1969 {797–3461}	1448 {653–3257}	1575 {653–3461}
	(15)	(23)	(18)	(30)	(26)	(24)	(34)	(58)
Mo	0.28 {0.06–0.52}	0.15 {0.05–0.94}	0.23 {0.05–0.55}	0.22 {0.05–0.55}	0.18 {0.05–0.94}	0.24 {0.07–0.55}	0.19 {0.05–0.94}	0.21 {0.05–0.94}
	(14)	(21)	(17)	(28)	(24)	(23)	(31)	(54)
Ni	5.79 {3.26–8.60}	4.01 {2.65–11.9}	4.27 {2.57–12.9}	4.82 {2.57–12.9}	4.08 {2.65–11.9}	5.25 {2.57–12.9}	4.11 {2.61–11.9}	4.30 {2.57–12.9}
	(15)	(23)	(18)	(30)	(26)	(24)	(34)	(58)
Pb	0.88 {0.60–1.17}	0.45 {0.27–2.92}	0.59 {0.27–1.24}	0.60 {0.27–1.24}	0.45 {0.27–2.92}	0.90 {0.27–1.25}	0.58 {0.27–2.92}	0.60 {0.27–2.92}
	(2)	(7)	(5)	(7)	(7)	(5)	(10)	(15)
Sb	0.90 {0.29–2.77}	1.04 {0.39–3.10}	1.17 {0.40–2.60}	0.91 {0.40–2.60}	1.21 {0.29–3.10}	0.89 {0.29–2.77}	0.94 {0.39–3.10}	0.92 {0.29–3.10}
	(14)	(17)	(17)	(28)	(20)	(21)	(29)	(50)
Se	0.76 {0.47–2.15}	0.86 {0.15–2.54}	0.71 {0.57–2.25}	0.71 {0.47–2.25}	0.90 {0.15–2.54}	0.73 {0.47–2.15}	0.90 {0.15–2.54}	0.81 {0.15–2.54}
	(8)	(17)	(11)	(18)	(18)	(12)	(26)	(38)
Zn	36.5 {24.3–68.7}	42.5 {31.3–135}	78.5 {55.2–105}	66.9 {27.4–103}	41.7 {24.3–135}	57.5 {24.3–135}	56.3 {27.2–110}	57.2 {24.3–135}
	(15)	(23)	(18)	(30)	(26)	(24)	(34)	(58)

AQL—above method qualification limit (the number of results exceeding AQL in round brackets); Range—as content {min–max}.

**Table 5 foods-10-02925-t005:** Extractable content (mg kg^−1^) of selected essential trace and potentially toxic elements in yerba mate *(Ilex paraguariensis)* in accordance with the whole population (including the percentage of extraction), origin, kind (type) and composition (purity).

	Origin			Kind (Type)		Composition (Purity)		Whole Population (*n* = 58)	
	Argentina (*n* = 15)	Brazil (*n* = 23)	Paraguay (*n* = 18)	Con Palo (*n* = 30)	Despalada (*n* = 26)	Pure Mate (*n* = 24)	With Additives (*n* = 34)	Content (mg kg^−1^)	% of Total Content
Elements	Median {Range} (AQL)	Median {Range} (AQL)	Median {Range} (AQL)	Median {Range} (AQL)	Median {Range} (AQL)	Median {Range} (AQL)	Median {Range} (AQL)	Median {Range} (AQL)	Median {Range} (AQL)
Al	66.2 {8.70–132}	51.1 {16.3–124}	41.6 {12.5–131}	47.6 {8.70–132}	54.4 {16.3–124}	60.1 {8.70–132}	49.2 {20.2–131}	52.9 {8.70–132}	28 {4–80}
	(15)	(23)	(18)	(30)	(26)	(24)	(34)	(58)	(58)
As	0.68 {0.62–0.74}	0.26 {0.24–0.28}	0.40 {0.18–0.54}	0.47 {0.18–0.62}	0.28 {0.24–0.74}	0.57 {0.40–0.74}	0.27 {0.18–0.62}	0.34 {0.18–0.74}	29 {20–66}
	(2)	(2)	(3)	(4)	(3)	(2)	(6)	(8)	(8)
Cd	0.05 {0.02–0.22}	0.06 {0.02–0.21}	0.11 {0.04–0.35}	0.07 {0.02–0.35}	0.06 {0.02–0.21}	0.05 {0.02–0.35}	0.08 {0.02–0.33}	0.07 {0.02–0.35}	17 {3–86}
	(13)	(20)	(13)	(23)	(23)	(21)	(27)	(48)	(48)
Co	0.24 {0.07–0.51}	0.18 {0.03–0.60}	0.16 {0.02–0.44}	0.18 {0.03–0.51}	0.17 {0.02–0.60}	0.16 {0.02–0.51}	0.19 {0.03–0.60}	0.18 {0.02–0.60}	37 {3–99.8}
	(14)	(17)	(15)	(26)	(20)	(22)	(25)	(47)	(47)
Cr	x	x	x	x	x	x	x	x	x
Cu	2.22 {0.43–4.52}	2.54 {0.77–4.42}	1.95 {0.56–4.91}	2.10 {0.43–4.91}	2.65 {0.77–4.42}	2.29 {0.43–4.06}	2.50 {0.86–5.79}	2.30 {0.43–5.79}	39 {8–99}
	(15)	(23)	(18)	(30)	(26)	(24)	(34)	(58)	(58)
Fe	13.8 {2.92–29.4}	11.5 {4.18–27.3}	13.4 {5.72–62.8}	13.8 {2.92–62.8}	11.9 {4.18–27.3}	12.9 {2.92–33.3}	13.8 {4.18–62.8}	13.2 {2.92–62.8}	8 {1–53}
	(15)	(23)	(18)	(30)	(26)	(24)	(34)	(58)	(58)
Hg	0.08 *	BQL	BQL	0.08 *	BQL	BQL	0.08 *	0.08 *	84 *
	(1)			(1)			(1)	(1)	(1)
Li	BQL	BQL	0.03 {0.01–0.04}	0.03 {0.01–0.04}	BQL	BQL	0.03 {0.01–0.04}	0.03 {0.01–0.04}	34 {33–35}
			(2)	(2)			(2)	(2)	(2)
Mn	513 {58.7–1110}	326 {126–860}	239 {29.3–913}	317 {29.3–1110}	342 {126–922}	376 {29.3–1110}	313 {63.9–1085}	343 {29.3–1110}	23 {2–66}
	(15)	(23)	(18)	(30)	(26)	(24)	(34)	(58)	(58)
Mo	BQL	0.14 *	0.11 *	0.11 *	0.14 *	0.11 *	0.13 {0.13–0.14}	0.13 {0.11–0.14}	57 {32–60}
		(1)	(1)	(1)	(1)	(1)	(2)	(3)	(3)
Ni	1.60 {0.77–2.67}	1.01 {0.08–2.35}	1.09 {0.29–2.70}	1.39 {0.29–2.70}	1.01 {0.08–2.67}	1.47 {0.29–2.67}	1.02 {0.08–2.70}	1.11 {0.08–2.70}	27 {3–70}
	(14)	(23)	(18)	(29)	(26)	(23)	(34)	(57)	(57)
Pb	0.82 *	0.71 {0.11–1.37}	0.44 {0.21–0.59}	0.51 {0.21–0.82}	0.71 {0.11–1.37}	0.76 {0.71–0.82}	0.44 {0.11–1.37}	0.59 {0.11–1.37}	56 {17–75}
	(1)	(3)	(3)	(4)	(3)	(2)	(5)	(7)	(7)
Sb	0.44 {0.13–0.80}	0.27 {0.09–0.65}	0.39 {0.08–1.17}	0.42 {0.08–1.17}	0.27 {0.09–0.65}	0.34 {0.08–0.68}	0.42 {0.09–1.17}	0.39 {0.08–1.17}	36 {9–94}
	(13)	(17)	(16)	(27)	(19)	(19)	(29)	(48)	(48)
Se	0.70 {0.15–1.25}	0.88 {0.83–0.95}	1.12 *	0.64 {0.15–1.12}	0.91 {0.83–1.25}	0.70 {0.15–1.25}	0.88 {0.18–1.12}	0.88 {0.15–1.25}	61 {19–88}
	(2)	(3)	(1)	(2)	(4)	(2)	(5)	(7)	(7)
Zn	16.1 {1.69–36.2}	19.7 {6.79–33.7}	23.9 {5.50–72.2}	19.5 {1.69–72.2}	19.4 {6.79–37.6}	15.4 {1.69–66.3}	21.6 {6.79–72.2}	19.4 {1.69–72.2}	35 {5–99}
	(15)	(23)	(18)	(30)	(26)	(24)	(34)	(58)	(58)

AQL—above method qualification limit (the number of results exceeding AQL in round brackets); Range—as content {min–max}; BQL—below quantification limit (if AQL = 0); *—the value (if AQL = 1); x—not determined due to interference, details in text.

**Table 6 foods-10-02925-t006:** Total content (as mean ± SD, mg kg^−1^) obtained in this study in comparison to the literature data.

Group		N	Al	As	Cd	Co	Cr	Cu	Fe	Hg	Li	Mn	Mo	Ni	Pb	Sb	Se	Zn	Ref.
Whole population		58	217 ± 67	1.35 ± 0.81	0.41 ± 0.13	0.49 ± 0.37	0.60 ± 0.28	6.69 ± 1.60	221 ± 168	0.33 ± 0.26	0.08 ± 0.05	1727 ± 735	0.26 ± 0.18	4.79 ± 2.00	0.86 ± 0.65	1.18 ± 0.68	0.93 ± 0.52	59.4 ± 24.9	This study
		54	361 ± 108	0.052 ± 0.251	0.410 ± 0.180	0.169 ± 0.956	0.528 ± 0.240	11.9 ± 2.06	205 ± 89.1	ND	0.085 ± 0.079	1078 ± 377	0.066 ± 0.325	2.74 ± 0.945	0.314 ± 0.181	ND	ND	63.6 ± 25.0	[11]
		32	90.4 ± 50.9	ND	0.19 ± 0.12	BQL	0.35 ± 0.13	5.17 ± 2.07	21.6 ± 16.5	ND	3.57 ± 1.94	66.4 ± 30.2	0.60 ± 0.40	1.39 ± 0.44	0.36 ± 0.41	ND	ND	32.5 ± 11.9	[23]
Origin	Argentina	15	214 ± 63	2.16 ± 0.36	0.32 ± 0.13	0.88 ± 0.43	0.73 ± 0.40	5.73 ± 1.57	216 ± 127	0.27 ± 0.16	0.05 ± 0.03	2543 ± 672	0.28 ± 0.14	5.71 ± 1.41	0.88 ^b^	1.07 ± 0.60	0.89 ± 0.49	43.1 ± 15.4	This study
		14	347 ± 60	0.04 ± 0.01	0.373 ± 0.167	0.209 ± 0.073	0.689 ± 0.18	12.6 ± 2.0	196 ± 42	ND	66.9 ± 22.4	1368 ± 256	0.051 ± 0.020	2.72 ± 0.718	0.222 ± 0.107	BDL	BDL	79.4 ± 17.7	[20]
		10	ND	ND	0.31 ^a^	ND	1.15 ^a^	7.72 ^a^	200 ^a^	ND	ND	1730 ^a^	ND	3.96 ^a^	0.40 ^a^	ND	ND	78.01 ^a^	[25]
	Brazil	23	216 ± 73	1.02 ± 0.80	0.44 ± 0.12	0.39 ± 0.28	0.61 ± 0.25	7.38 ± 1.22	203 ± 225	0.37 ± 0.36	0.06 ± 0.04	1525 ± 448	0.22 ± 0.21	4.38 ± 1.98	0.92 ± 0.88	1.28 ± 0.78	0.91 ± 0.54	55.9 ± 27.4	This study
		19	291 ± 56	0.05 ± 0.03	0.491 ± 0.225	0.121 ± 0.094	0.37 ± 0.19	11.4 ± 2.1	154 ± 48	ND	74.5 ± 134	987 ± 352	0.066 ± 0.040	2.38 ± 1.14	0.407 ± 0.230	BDL	BDL	44.2 ± 14.7	[20]
		9	378 ± 113	0.04 ± 0.02	0.40 ± 0.13	0.21 ± 0.09	ND	9.22 ± 0.95	280 ± 221	ND	0.11 ± 0.06	1313 ± 592	0.05 ± 0.05	2.19 ± 0.55	0.28 ± 0.12	ND	0.03 ± 0.02	55 ± 13	[22]
	Paraguay	18	219 ± 63	1.59 ± 0.67	0.45 ± 0.10	0.33 ± 0.11	0.52 ± 0.11	6.44 ± 1.62	257 ± 102	0.34 ± 0.16	0.10 ± 0.05	1224 ± 339	0.28 ± 0.17	4.53 ± 2.25	0.78 ± 0.38	1.23 ± 0.62	0.97 ± 0.55	79.0 ± 12.7	This study
		14	384 ± 62	0.06 ± 0.03	0.295 ± 0.082	0.101 ± 0.084	0.70 ± 0.13	11.1 ± 1.9	226 ± 122	ND	59.1 ± 33.3	730 ± 150	0.089 ± 0.022	2.81 ± 0.720	0.314 ± 0.178	BDL	BDL	77.3 ± 25.4	[20]
		5	ND	ND	0.30 ^a^	ND	0.88 ^a^	7.28 ^a^	130 ^a^	ND	ND	680 ^a^	ND	3.03 ^a^	0.45 ^a^	ND	ND	115.05 ^a^	[25]
Kind (type)	con palo	30	216 ± 63	1.70 ± 0.64	0.40 ± 0.13	0.56 ± 0.42	0.62 ± 0.31	6.26 ± 1.61	237 ± 118	0.32 ± 0.16	0.08 ± 0.06	1776 ± 836	0.26 ± 0.15	5.06 ± 2.06	0.81 ± 0.36	1.09 ± 0.50	0.87 ± 0.46	63.7 ± 20.3	This study
		10	ND	ND	0.33 ^a^	ND	0.95 ^a^	7.35 ^a^	150 ^a^	ND	ND	1070 ^a^	ND	3.27 ^a^	0.39 ^a^	ND	ND	96.49 ^a^	[25]
	despalada	26	217 ± 72	1.15 ± 0.86	0.43 ± 0.12	0.43 ± 0.29	0.61 ± 0.23	7.07 ± 1.49	208 ± 215	0.35 ± 0.36	0.06 ± 0.03	1615 ± 541	0.25 ± 0.22	4.47 ± 1.93	0.92 ± 0.88	1.36 ± 0.85	0.97 ± 0.60	55.5 ± 28.7	This study
		5	ND	ND	0.28 ^a^	ND	1.29 ^a^	8.03 ^a^	220 ^a^	ND	ND	1990 ^a^	ND	4.41 ^a^	0.52 ^a^	ND	ND	78.10 ^a^	[25]
Composition (purity)	pure mate	24	235 ± 69	1.79 ± 0.83	0.38 ± 0.14	0.65 ± 0.44	0.62 ± 0.19	6.15 ± 1.59	250 ± 191	0.29 ± 0.13	0.07 ± 0.05	2037 ± 837	0.27 ± 0.15	5.38 ± 2.18	0.84 ± 0.36	1.11 ± 0.61	0.83 ± 0.42	58.2 ± 27.4	This study
	with additives	34	205 ± 62	1.24 ± 0.77	0.43 ± 0.13	0.38 ± 0.26	0.59 ± 0.33	7.06 ± 1.50	201 ± 146	0.37 ± 0.32	0.08 ± 0.05	1509 ± 558	0.24 ± 0.20	4.37 ± 1.74	0.88 ± 0.76	1.24 ± 0.71	0.97 ± 0.56	60.3 ± 22.9	This study

N—number of samples; BQL—below quantification limit; BDL—below detection limit; SD—standard deviation; ND—not determined; ^a^—mean (if SD was not reported); ^b^—median (if *n* < 3).

## Data Availability

All data presented in the article are available from the corresponding author.

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
