# Peer review of "Elemental and Speciation Analyses of Different Brands of Yerba Mate (Ilex paraguariensis)"

_foods, 2021, doi:10.3390/foods10122925_

Round 1

Reviewer 1 Report

This paper reports data on As, Fe, and other trace elements in yerba mate. I think the study is a good choice and data are worthy. However, the following improvements are needed:

  1. The title of the paper is not well organized, ther are too many "and", and the mean it expressed is not clear.
  2.  The abstract is now well organized, and results and conclusions related to purposes and new finding of the study are hardly found.
  3.  The Part of Introduction includes too many contents that are relative too many aspects which are less relative to the study. I think the most important for revision is to make this papre have a clear distination.
  4.  As it is not clear for purpose of this paper, and Parts of Results and Discussion of this paper were all not well organized. I am unable to know what kind of problem the author hope to solve by this paper with a lot of analysing data.

Author Response

Reviewer 1 comments

This paper reports data on As, Fe, and other trace elements in yerba mate. I think the study is a good choice and data are worthy. However, the following improvements are needed:

Thank you for reviewing the article and suggested corrections. All of them were taken into account as indicated by the reviewer.

The title of the paper is not well organized, ther are too many "and", and the mean it expressed is not clear.

The title has been changed to:

Arsenic and iron species in the context of selected elements concentration in different brands of yerba mate (Ilex paraguariensis)

The abstract is now well organized, and results and conclusions related to purposes and new finding of the study are hardly found.

Abstract has been reorganized in the new shape:

ABSTRACT: In this work, a methodology for determination of As(III), As(V), DMA, Fe(II) and Fe(III) in fifty–eight samples (forty–nine products of thirteen brands from three countries) commercial yerba mate (Ilex paraguariensis) was performed. The hyphenated technique: high performance liquid chromatography inductively coupled plasma optical emission spectrometry (HPLC–ICP OES) was used. Arsenic was determined below quantification limit in 38 samples of yerba mate. As(III) was found at the level 0.09 and 0.08 mg kg–1. The As(V) content was in the range: 0.21 to 0.28 mg kg–1. The content of DMA was found the highest of the three arsenic species in the range: 0.21 to 0.47 mg kg–1. The content of Fe(II) and Fe(III) was found in the range: 0.61 to 15.4 mg kg–1 and 0.66 to 43.1 mg kg–1, respectively and the dominance of Fe(III) was observed. Moreover, total and extractable content of 16 elements were determined The results have been subjected to statistical analysis in order to establish relationships between samples of the same origin (country), kind (type) and composition (purity).

The Part of Introduction includes too many contents that are relative too many aspects which are less relative to the study. I think the most important for revision is to make this papre have a clear distination.

Introduction part has been essentially changed:

Yerba mate (Ilex paraguariensis St. Hil.) is a native South American tree. It is an important commercial product, consumed in the largest quantities in Brazil and Uruguay, while Argentina is the largest exporter [1]. The consumption of yerba mate has been expanded to different countries e.eg.: Spain, France, Italy, Germany, Korea, Japan, Syria, Russia, United States and Australia [2]. Yerba mate can accumulate both essential trace and potentially toxic elements (PTEs) [1] and their content depends on certain factors, such as soil type, exposure of plants to pollution, even harvest season [3]. High tolerance to metals or metalloids has evolved in a number of plant species. Tolerant plants are often excluders, limiting the entry and translocation, or rarely hyperaccumulators combines extremely high tolerance to, and foliar accumulation of, trace elements [4].

Nowadays (2021), the determination of the total concentration of trace elements in yerba mate (Ilex paraguariensis) or its tea is not enough to evaluate the nutritional or toxic potential of the product. In infusions and decoctions, metals and metalloids may exist as either simple or complexed ions. This fact can affect the bioavailability of elements by humans [5]. What is more, nutritional and toxic potential depends from element species, therefore a speciation analysis of elements, such as As, Cr, Fe, Se is crucial. The toxicity of elements forms is significantly different, and the greatest obstacle to the efficient determination of these forms is the ease of converting from one form to another [6, 7]. In opposite to arsenic, iron (Fe) is an essential trace and the fourth most abundant element in the Earth’s crust. Two main species, Fe(II) and Fe(III), are thermodynamically stable and kinetically reactive, however the role and the demand of these forms in living organisms are different [8]. Due to ambiguous classification of this material in the literature (as tea [9], laboratory plant [10] or wild–growing plant [11]), yerba mate (Ilex paraguariensis) is used willingly by authors as an application material for new analytical methods [1, 5, 12–17]. It is surprising that the content of essential trace and potentially toxic elements in yerba mate have been obtained by ICP OES [18], ICP MS [1, 19] or both ICP OES and ICP MS [11, 20], excluding any speciation studies. Admittedly, some methods for determining the species of selenium [14], arsenic [12], iron [13], as well as iron and arsenic [17] in several samples of yerba mate have already been presented. However, speciation studies have not yet been carried out on a larger number of samples.

In this study, to investigate the speciation of arsenic and iron in yerba mate (Ilex paraguariensis), 58 samples were collected from Polish market. Additionally, the determination of selected essential trace and potentially toxic elements (PTEs), i.e. Al, As, Cd, Co, Cr, Cu, Fe, Hg, Li, Mn, Mo, Ni, Pb, Sb, Se, and Zn, was performed using ICP OES. The determination of As(III), As(V) and DMA as well as Fe(II) and Fe(III) in yerba mate is a novel, although it is an enlargement of the preliminary studies, conducted for the first usage of HPLC–HG–ICP OES with MSIS as an interface [12], the first comparison of HPLC–ICP OES and HPLC–ICP OES [13], and the first combination of two HPLC systems with ICP OES through MSIS (2 HPLC–MSIS–ICP OES) [17].

As it is not clear for purpose of this paper, and Parts of Results and Discussion of this paper were all not well organized. I am unable to know what kind of problem the author hope to solve by this paper with a lot of analysing data.

Since the aim of the study is the speciation studies of arsenic and iron, we decided to discuss it firstly. Next, we presented the results of the ICP OES analysis in order to compare the results of the speciation analysis with the total and extractable content of these and other selected elements. In the course of the research, the results of ICP OES turned out to be scientifically significant, therefore they were also exposed, and in the further part of the article statistically analyzed. In our opinion, the structure seemed clear, however the text has been changed to better indicate the purpose of the analysis. Additionally, the Conclusion section has been shortened to one paragraph:

Results of arsenic and iron speciation studies were comparable with obtained in our previous studies [12, 13, 17], occurring selected As and Fe species in yerba mate as well as occasionally the high content, are accidental rather than related to studied factors, i.e. country of origin, kind, composition or packing. However, the obtained results suggest that the speciation studies of selected essential trace and potentially toxic elements could give some information about the origin, type and composition of yerba mate. Nevertheless, new methods of multi–element speciation analysis as well as an appropriate preparation procedure of this sample matrix should be developed for this purpose. Since the non–extractable content predominated in total iron and arsenic content, it is presumed that the elements were built into the plant tissues during the growth stage. This indicates that the determination of the arsenic and iron grades as well as extractable content must be assessed in the context of the total content of these elements. In the case of the total content of selected essential trace and potentially toxic elements, no significant differences were noticed between the yerba mate products packed in Poland and those packed in the countries of South America. In turn, the correlations between the elements were rather moderate, however also consistent with the literature. Probably the soil, which was used for the cultivation of Ilex paraguariensis, is the main factor influencing the elemental composition, and not the country of origin as previously thought.

Reviewer 2 Report

The manuscript reported the analysis of potentially toxic elements from yerba mate. The obtained data were statistically studied and interpreted. Some minor points are required to be improved.

  • Line 9: full expression of DMA should be presented before the abbreviation.
  • The introduction should be rewritten concisely. The current introduction is hard to follow the main story.
  • Line 181: it should be "was summarized in Table 2."
  • Part 2.6 should be moved to the results section.
  • The conclusion should be within 1 paragraph and report the main findings of the study only.

Author Response

Reviewer 2 comments

The manuscript reported the analysis of potentially toxic elements from yerba mate. The obtained data were statistically studied and interpreted. Some minor points are required to be improved.

We would like to thank you for the review. All proposed changes have been introduced in the text.

  • Line 9: full expression of DMA should be presented before the abbreviation.

The full expression: dimethylarsinic acid (DMA) has been introduced.

  • The introduction should be rewritten concisely. The current introduction is hard to follow the main story.

Introduction part has been essentially changed:

  • Line 181: it should be "was summarized in Table 2."

The sentence has been changed to:

Full experimental conditions of HPLC–HG–ICP OES were summarized in Table 2.

  • Part 2.6 should be moved to the results section.

Paragraph 2.6. contains technical and metrological information on measurements and validation of the analytical procedure. The paragraph does not present the results of sample analyzes. Due to this, we have decided to keep the paragraph in part 2. Materials and methods. We hope this will be accepted by the reviewer. To avoid ambiguity, we changed the title of the paragraph:

2.6. ICP OES determination of selected elements content

  • The conclusion should be within 1 paragraph and report the main findings of the study only.

The Conclusion section has been shortened to one paragraph:

Results of arsenic and iron speciation studies were comparable with obtained in our previous studies [12, 13, 17], occurring selected As and Fe species in yerba mate as well as occasionally the high content, are accidental rather than related to studied factors, i.e. country of origin, kind, composition or packing. However, the obtained results suggest that the speciation studies of selected essential trace and potentially toxic elements could give some information about the origin, type and composition of yerba mate. Nevertheless, new methods of multi–element speciation analysis as well as an appropriate preparation procedure of this sample matrix should be developed for this purpose. Since the non–extractable content predominated in total iron and arsenic content, it is presumed that the elements were built into the plant tissues during the growth stage. This indicates that the determination of the arsenic and iron grades as well as extractable content must be assessed in the context of the total content of these elements. In the case of the total content of selected essential trace and potentially toxic elements, no significant differences were noticed between the yerba mate products packed in Poland and those packed in the countries of South America. In turn, the correlations between the elements were rather moderate, however also consistent with the literature. Probably the soil, which was used for the cultivation of Ilex paraguariensis, is the main factor influencing the elemental composition, and not the country of origin as previously thought.

Round 2

Reviewer 1 Report

Quality of  the manuscript has been improved this time, however I am not happy with the title that can express less mean of the study. Also, conclusions of the paper were not summarized as well as it should be.  

Author Response

Reviewer 1 comments:

Thank you for reviewing the article and suggested corrections.

Quality of the manuscript has been improved this time, however I am not happy with the title that can express less mean of the study.

The title has been changed:

Elemental and speciation analyses of different brands of yerba mate (Ilex paraguariensis)

Also, conclusions of the paper were not summarized as well as it should be.

The paragraph has been changed:

New methods of single–element speciation analysis have been applied for the first time to 58 samples of commercial yerba mate. As expected, the results of arsenic and iron species were comparable with those obtained for several samples in our preliminary studies [12, 13, 17]. By comparing the content of selected essential trace and potentially toxic elements obtained with two different procedures of sample preparation (digestion, extraction), it was shown that non–extractable content predominated in yerba mate, especially for iron (approx. 92%). According to this, the determination of the arsenic and iron species as well as extractable content must be assessed in the context of the total content of these elements since the elements were built into the plant tissues during the growth stage. The occurrence of selected As and Fe species in yerba mate as well as occasionally the high content, are accidental rather than related to studied factors, i.e. country of origin, kind, composition or packing. What is more, no significant differences were noticed between the yerba mate products packed in Poland and those packed in the countries of South America. Probably the soil, which was used for the cultivation of Ilex paraguariensis, is the main factor influencing the elemental composition, and not the country of origin as previously thought. In the future perspective, new methods of multi–element speciation analysis as well as an appropriate preparation procedure of this sample matrix should be developed to obtain more information about the origin, type and composition of yerba mate.